# Correlation between Preoperative Coronary Artery Stenosis Severity Measured by Instantaneous Wave-Free Ratio and Intraoperative Transit Time Flow Measurement of Attached Grafts

**DOI:** 10.3390/medicina56120714

**Published:** 2020-12-18

**Authors:** Almas Tolegenuly, Rasa Ordiene, Arslan Mamedov, Ramunas Unikas, Rimantas Benetis

**Affiliations:** 1Department of Cardiac, Thoracic and Vascular Surgery, Hospital of Lithuanian University of Health Sciences Kauno Klinikos, Medical Academy, Lithuanian University of Health Sciences, Eivenių 2, LT-50009 Kaunas, Lithuania; zolotoyarslan2@gmail.com (A.M.); benetis@lsmuni.lt (R.B.); 2Department of Cardiology, Medical Academy, Lithuanian University of Health Sciences, Eivenių 2, LT-50009 Kaunas, Lithuania; rasa.ordiene@lsmuni.lt (R.O.); unikas@lsmuni.lt (R.U.)

**Keywords:** coronary artery bypass grafting, instantaneous wave-free ratio, transit time flow measurement, competitive flow, early graft failure

## Abstract

*Background and Objectives:* To assess the correlation between the degree of target coronary artery stenosis measured by instantaneous wave-free ratio (iFR) and the intraoperative transit time flow measurement (TTFM) of attached grafts as well as evaluate flow competition between the native coronary artery and the attached graft according to the severity of stenosis. *Materials and Methods:* In total, 89 grafts were subjected to intraoperative transit time flow measurement after coronary artery bypass grafting (CABG) in 25 patients with multivessel coronary artery disease (CAD). The iFR was evaluated for all coronary arteries with grafts. The coronary artery stenoses were divided into three groups based on the iFR value: iFR < 0.86 (group 1); iFR 0.86–0.90 (group 2); and iFR > 0.90 (group 3). *Results:* The mean graft flow (MGF) was 46.9 ± 18.4 mL/min for group 1, 45.3 ± 20.9 mL/min for group 2, and 31.3 ± 18.5 mL/min for group 3. A statistically significant difference was confirmed between groups 1 and 3 (*p* = 0.002) and between groups 2 and 3 (*p* = 0.025). The pulsatility index (PI) was 2.49 ± 1.20 for group 1, 2.66 ± 2.13 for group 2, and 4.70 ± 3.66 for group 3. A statistically significant difference was found between groups 1 and 3 (*p* = 0.006) and between groups 2 and 3 (*p* = 0.032). Backward flow was detected in 7.5% of grafts for group 1, in 16.6% of grafts for group 2, and in 16% of grafts for group 3. A statistically significant difference was found between groups 1 and 2 (*p* = 0.025) and between groups 1 and 3 (*p* = 0.029). *Conclusions:* The iFR is a useful tool for predicting the impact of competitive flow observed between a native artery and an attached graft. The effect of competitive flow significantly increases when the graft is attached to a vessel with mild coronary stenosis. In a coronary artery where the iFR was not hemodynamically significant, the MGF was lower, the PI was higher, and a larger proportion of grafts with backward flow (BF) was detected compared to when there was significant stenosis (iFR < 0.86).

## 1. Introduction

Competitive flow from the native coronary artery to an attached graft is a powerful factor causing early graft failure [1]. However, competitive flow can be avoided by through detailed selection of the patients, required graft position, and target artery to bypass [2]. Consequently, physiology-based revascularization has become an essential part of the evidence-based management of coronary artery disease (CAD) patients [3]. The instantaneous wave-free ratio (iFR) is an IA class indication for the evaluation of intermediate coronary stenosis and a guide to indications for revascularization [4,5]. Among the various existing methods for the evaluation of coronary physiology, significant advantages of iFR have been demonstrated [6].

DEFINE-FLAIR and iFR-SWEDEHEART are randomized and prospective trials which compared the use of fractional flow reserve (FFR) and iFR in guiding revascularization strategy, and both showed similar and comparable results [7,8]. Given the close association between iFR and flow in the coronary artery, iFR may be suitable for risk stratification and to determine a management strategy [6]. Transit time flow measurement is the tool most commonly used for intraoperative graft quality control after the coronary artery bypass grafting (CABG) procedure [9] and, presently, no comparisons between transit time flow measurement (TTFM) and iFR data for CABG patients have been published so far.

One study, published by Honda et al. [10], used TTFM to assess the competitive flow of arterial grafts for FFR-guided CABG patients and showed good correlation between the coronary artery lesion severity evaluated by FFR and the incidence rate of flow competition between the native coronary artery and the attached graft. The aim of our study is to evaluate the correlation between intraoperative graft flow measurements and the iFR-determined coronary lesion severity to assess the potential of competitive flow as a factor for predicting early graft failure.

## 2. Materials and Methods

### 2.1. Study Groups

This prospective study included 25 multivessel stable CAD patients who underwent 89 intraoperative graft assessments using transit time flow measurements after CABG surgery (Figure 1). The enrollment of the patients was consecutive. The iFR was measured for all angiographically intermediate (40–75% by diameter) stenoses of coronary arteries. The grafts were divided into three groups according to the preoperative lesion severity: group 1 (iFR < 0.86), group 2 (iFR 0.86–0.90), and group 3 (iFR > 0.90). The grafts in this study were attached despite negative iFR since the consensus to graft based on angiography findings was made before iFR measurements. Graft flow was assessed using TTFM based on four variables: mean graft flow (MGF), pulsatility index (PI), backward flow (BF), and diastolic filling % (DF%).

All the patients were on a standard treatment according to European Society of Cardiology (ESC) guidelines on chronic coronary syndrome [11].

Permission for the study was confirmed by the Kaunas Regional Biomedical Research Ethics Committee on 9 September 2019 (Nr. BE-2-70).

### 2.2. iFR Measurement

iFR is a functional assessment of stenosis that can be performed by measuring the intracoronary flow in the catheterization laboratory and calculated as the mean pressure distal to the stenosis during the diastolic wave-free period (Pd wave-free period) divided by the mean aortic pressure during the diastolic wave-free period (Pa wave-free period) [4]. Physiological measurements were performed in the standard manner using a coronary pressure guidewire (Verrata, Philips Volcano, San Diego, CA, USA). Before every measurement, intracoronary nitrates were administered to avoid vasomotor reactions. The iFR cutoff point was 0.90, where stenosis with iFR > 0.9 was considered hemodynamically nonsignificant [4,12], while stenosis when iFR was 0.86–0.90 was the so-called “gray zone” and iFR < 0.86 was considered severe coronary stenosis [13]. In our study all physiology measurements of the coronary artery were done by the same operator and in all targeted vessels iFR was measured distally and with pullback to localize the most severe lesion.

### 2.3. Revascularization and Graft Flow Measurement

All CABG procedures were performed via median sternotomy and cardiopulmonary bypass (CPB) with heparinization of 300 international units/kg and ensured activated clotting time (ACT) > 480 s. For intraoperative graft flow assessment, we used the Medistim VeriQ Cardiac (VeriQ model VQ4122C, Oslo, Norway) apparatus. Internal mammary artery (IMA) grafts required skeletonization of the small segment to improve fitting. The sequential grafts were evaluated separately by clamp-on of neighbor distal anastomosis and for ease of calculation were defined as separate grafts. The blood flow was evaluated after performing intraoperative angiography, and proximal anastomosis was attached with adequate de-airing. Measurement of all grafts was possible using a wide range of probe sizes; we typically used 3, 4, and 5 mm probes.

**Mean graft flow.** MGF is an indicator for assessing bypass flow and is represented in mL/min. The graft quality, flow through the native coronary artery, distal vascular bed, and arterial pressure may impact MGF. During synchronization of the MGF with electrocardiography (ECG), systolic (red color) and diastolic (blue color) filling was recorded on a display (Figure 2). The recommended value of good flow for the IMA grafts is >20 mL/min, while for saphenous vein grafts (SVG), it is >40 mL/min. Grafts with values less than 5 mL/min are considered poor flow [14,15].

**Pulsatility index**. Graft flow resistance can be estimated by the pulsatility index and is represented as an absolute number. PI is the numeric difference between the maximum flow (*Q max*) and the minimum flow (*Q min*) divided by the mean flow (*Q mean*), which provides information on flow patterns (Figure 2). The formula is
PI = [(*Qmax* − *Qmin)/Qmean*].

The recommended cutoff values range from 1 to 5. Values > 5 are considered to indicate unsatisfactory graft flow [15,16].

**Backward flow.** BF indicates flow competition between the native coronary artery and the graft. BF expresses the percentage of graft blood flow redirected to the graft and is measured during one complete cardiac cycle. If the percentage of the reverse flow area is more than 3% of total flow, this is considered to be a positive value of BF [14,17]. The BF is registered as the area below the zero line (Figure 2).

**Diastolic filling %.** DF% expresses the proportion of diastolic graft flow during the entire graft flow (Figure 2). The DF% is calculated using the formula
DF% = [(Qdiastole/Qsystole + Qdiastole)].

The total flow in the diastole should exceed 50% of the MGF, and proportions < 25% are considered to indicate unsatisfactory diastolic filling [14,18].

### 2.4. Intraoperative Angiography

All CABG cases were performed in a hybrid surgery room. We used the Siemens Artis Zeego multi-axis system (Munich, Germany) for angiography of the attached grafts. Graft flow assessment was done after the angiographic control and revisions or reinterventions of angiographic defects were done.

### 2.5. Statistical Analysis

Statistical analysis was performed using software IBM SPSS Statistics 23 (Armonk, NY, USA: IBM Corp. Software). There were defined statistical characteristics such as the total observation number, mean, median, and standard deviation using descriptive statistics. Continuous variables are presented as the mean (standard deviation (SD)) and as the median (interquartile range (IQR)), while categorical data are expressed as numbers (percentages). After testing for normality, group differences were tested using Student’s *t*-test and Kruskal–Wallis analysis of variance to compare samples. The correlation between the quantitative TTFM data and iFR was evaluated by one-way ANOVA analysis and Spearman correlation coefficient analysis. The qualitative analysis in the groups was evaluated using chi-square tests. Variables with a two-sided *p* value < 0.05 were considered statistically significant.

## 3. Results

In total, 25 patients (age range of 48–78 years; mean 63.8 ± 8.9 years) participated in the study. Table 1 shows the patient characteristics. A total of 25 arterial grafts and 64 vein grafts were included in our study. For the left coronary artery territory, the left internal mammary artery (LIMA) was used in situ. Right internal mammary artery (RIMA) and bilateral internal mammary artery (BIMA) were not used in this study. Saphenous vein grafts (SVG) were mainly used for the right coronary artery or circumflex to graft (Table 2). Perioperative mortality was documented for one patient. For the other patients, the postoperative period was related to complications, such as cardiogenic shock (*n* = 2), respiratory failure (*n* = 1), and arrhythmia in the form of ventricular fibrillation (*n* = 1). Only one graft defect (1.12%) was detected by TTFM (after angiography) and required reintervention.

### Graft Flow Assessment

A total of 89 measurements were performed by TTFM.

There were 40 grafts (29 SVG/11 LIMA) in group 1 (iFR < 0.86), 24 grafts (17 SVG/7 LIMA) in group 2 (iFR 0.86–0.90), and 25 grafts (18 SVG/7 LIMA) in group 3 (iFR > 0.90).

The mean graft flow was 46.9 ± 18.4 mL/min (range 12–78 mL/min) for group 1, 45.3 ± 20.9 mL/min (range 14–72 mL/min) for group 2, and 31.3 ± 18.5 mL/min (range 16–75 mL/min) for group 3 (Figure 3). Statistically significant differences were found between groups 1 and 3 (*p* = 0.002) and between groups 2 and 3 (*p* = 0.025).

The pulsatility index was 2.49 ± 1.20 (range 0.8–9.7) for group 1, 2.66 ± 2.13 (range 0.7–7) for group 2, and 4.70 ± 3.66 (range 1.1–9.1) for group 3 (Figure 4). Statistically significant differences were found between groups 1 and 3 (*p* = 0.006) and between groups 2 and 3 (*p* = 0.032).

Backward flow was detected in 3 grafts (7.5%) for group 1, in 4 grafts (16.6%) for group 2, and in 4 grafts (16%) for group 3 (Figure 5). Statistically significant differences were found between groups 1 and 2 (*p* = 0.025) and between groups 1 and 3 (*p* = 0.029). No significant differences were found between groups 2 and 3 (*p* = 0.195).

The diastolic filling % was 76.3 ± 12.4% (range 40–83%) for group 1, 73.5 ±10.1% (range 45–82%) for group 2, and 70.7 ± 11.9% (range 41–84%) for group 3. Statistically nonsignificant differences were found between all groups: between groups 1 and 3 (*p* = 0.175), between groups 2 and 3 (*p* = 0.351), and between groups 1 and 3 (*p* = 0.175).

The correlation coefficient between iFR and MGF was −0.372 (*p* = 0.024), and between iFR and PI it was 0.428 (*p* = 0.044) in all grafts. In venous grafts, separately, the correlation coefficient between iFR and MGF was −0.330 (*p* = 0.064), and between iFR and PI it was 0.275 (*p* = 0.091). In arterial grafts, the coefficient of correlation between iFR and MGF was −0.460 (*p* = 0.048) and between iFR and PI it was 0.563 (*p* = 0.002).

During LIMA-left anterior descending artery (LAD) graft angiography, reverse contrast flow was noted in two grafts (2.2%). Preoperative iFR values of the mentioned LAD arteries were 0.91 and 0.93. In all cases, BF in grafts was confirmed using TTFM (Figure 6).

## 4. Discussion

Mechanical myocardial revascularization—percutaneous coronary intervention (PCI) and coronary artery bypass grafting (CABG), in addition to guideline-based medical therapy, remain the mainstay in the treatment of symptomatic stable CAD patients [3,5]. Unlike PCI, where the flow is restored by opening the lumen of the target coronary artery, additional flow is created in the native coronary artery distal to stenosis during the CABG. There is a competitive flow between the native artery and the attached graft, which depends on the degree of stenosis of the native artery. The early (one week) clinical records and angiograms detected a 6.5% incidence of competitive flow in the investigated arterial grafts [2]. In a large US cohort of 500,154 patients who underwent revascularization, 12% of the interventions were classified as inappropriate [19]. This indicates that with overdiagnosis and unreasonable CABG, the impact of competitive flow from the native coronary artery to graft significantly increases.

A correlation between TTFM and FFR has been described by Honda et al. [10] in which the impact of the severity of stenosis on competitive flow was evident [10]. In our study, we confirmed that with increasing stenosis of the native coronary artery, MGF increased, PI decreased, and the proportion of grafts with BF decreased. We can infer that iFR allows us to predict a certain level of risk due to competitive flow. Although there was a weak negative correlation between iFR and MGF, as well as a weak positive correlation between iFR and PI in all grafts, single arterial grafts showed a stronger correlation in these parameters.

Unlike in our study, the study by Honda et al. [10] in which a correlation was found between FFR and TTFM valuated only internal mammary artery grafts. Despite a different cutoff value for MGF of IMA grafts at 20 and 40 mL/min for SVG [14], we included all grafts (venous and arterial) in the investigation. Interestingly, the percentage of venous and arterial grafts in all three groups were equally distributed.

Earlier research reported about the “survivor” of IMA grafts despite competitive flow for the native coronary artery, with observed regression of LAD lesions (less than 25%) three and one years after surgery, and the anatomical patent IMA grafts were angiographically occluded [20]. Angiographic patency of IMA was restored in the occluded native coronary artery flow using a coronary angioplasty balloon. This could also be the reason for the long-term survival of in situ IMA grafts (LIMA 96.4%, RIMA 88.2% patency rate) while SVG patency was only 61% in the 15-year follow-up [21].

In a later study, reopening of the graft lumen associated with the progression of native coronary artery stenosis was not observed [2]. The radial artery (RA) is a popular second conduit for coronary bypass grafting. In several studies, RA grafts were found to have superior patency compared to SVG; however, this is not recommended in the presence of moderate stenosis (<90%) as it is prone to spasms [22,23,24]. Surgeons may prefer to use the gastroepiploic artery (GEA). The ten-year patency of the GEA was approximately 80% [25]. 

The main disadvantages of using the GEA is vascular spasms and a high sensitivity to competitive flow through the native artery [26,27]. As we know that competitive flow is a predictor of early arterial graft failure, some authors recommend SVG for bypassing the right coronary artery (RCA) with moderate stenosis [1]. The SVG patency rate was not sensitive to target vessel degree of stenosis as it was perfused directly from the aorta with higher pressures compared with arterial grafts [28]. This was a pilot study where we evaluated the impact of the native coronary artery stenosis measured by iFR on graft flow regardless of graft type—arterial or venous. We can assume that the presence of a significant competitive flow indicates the presence of a sufficient flow from the native artery and an overdiagnosed level of stenosis in the native coronary artery.

Due to the initial assessment of the graft using angiography, it was possible to exclude the surgeon’s technical errors: anastomosis quality, graft defects, and target vessel errors. There was one defect (1.12%) that required reintervention after TTFM, and the incidence of technical aspects was significantly lower than reported in previous studies (approximately 3% out of all grafts) [9,29].

### Limitations

Our research is limited by the relatively small number of patients and as a single-center study. For the calculations, we used data from both arterial and venous grafts. There was also no separation of the revascularization area. As there are anatomical and physiological differences, this can produce variation in the data. This study combines the cardiologists’ and surgeons’ efforts, and recruiting a large sample of patients within a certain period of time was a limitation of our study. The use of sequential grafts prolonged the time of TTFM as this required more manipulations in the chest. The mid-term (3–6-month) computed tomography (CT) angiography follow-up is in progress.

## 5. Conclusions

The iFR is a useful tool to predicting the impact of competitive flow seen between the native artery and an attached graft. The effect of competitive flow significantly increases when the graft is attached to a vessel with mild coronary stenosis. In a coronary artery where the iFR was not hemodynamically significant, the MGF was lower, the PI was higher, and a larger proportion of grafts with BF was detected compared to when there was significant stenosis (iFR < 0.86).

## Figures and Tables

**Figure 1 medicina-56-00714-f001:**
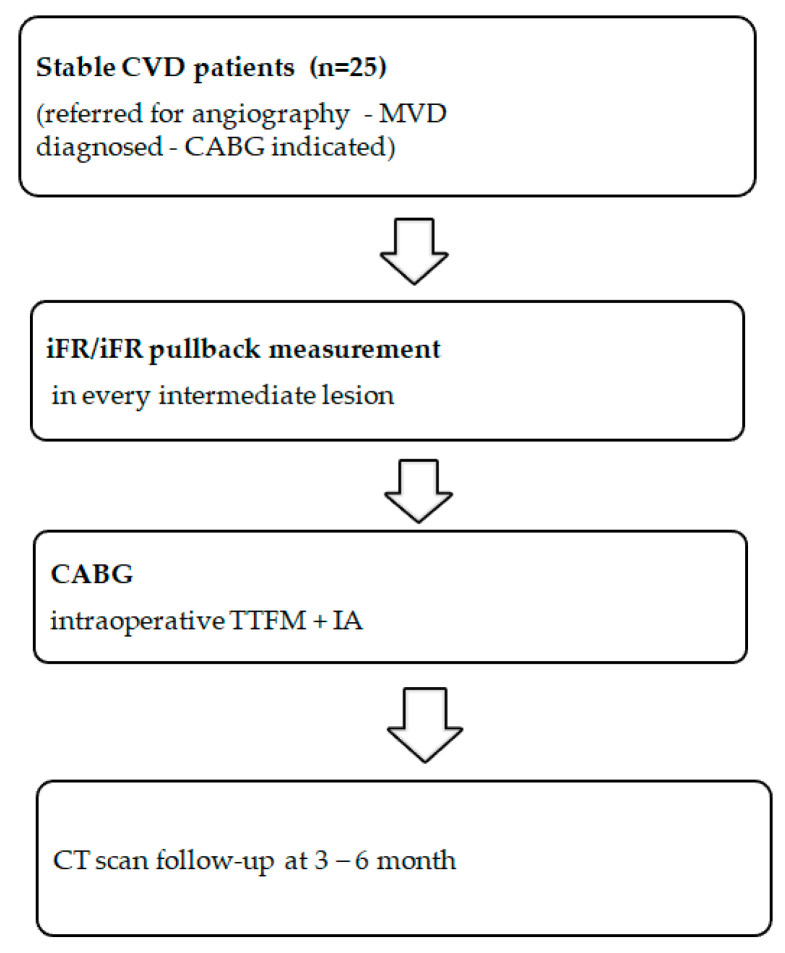
Flow chart of the study. CVD—coronary vessel disease; MVD—multivessel disease; iFR—instantaneous wave-free ratio; CABG—coronary artery bypass grafting; TTFM—transit time flow measurement; IA—intraoperative angiography; CT—computed tomography.

**Figure 2 medicina-56-00714-f002:**
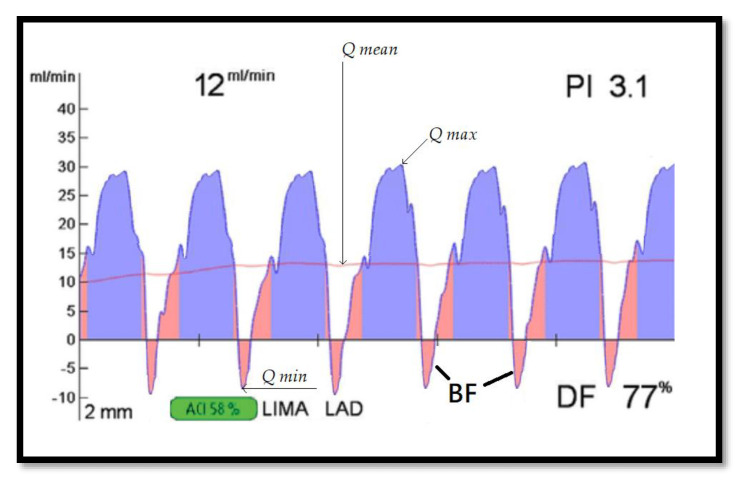
Graft flow assessment of the left internal mammary artery (LIMA) graft to the left anterior descending artery (LAD); 12 mL/min—mean graft flow; PI—pulsatility index; BF—backward flow; DF—diastolic filling; ACI—acoustic coupling index; *Q max*—maximum flow; *Q min*—minimum flow; and *Q mean*—mean flow.

**Figure 3 medicina-56-00714-f003:**
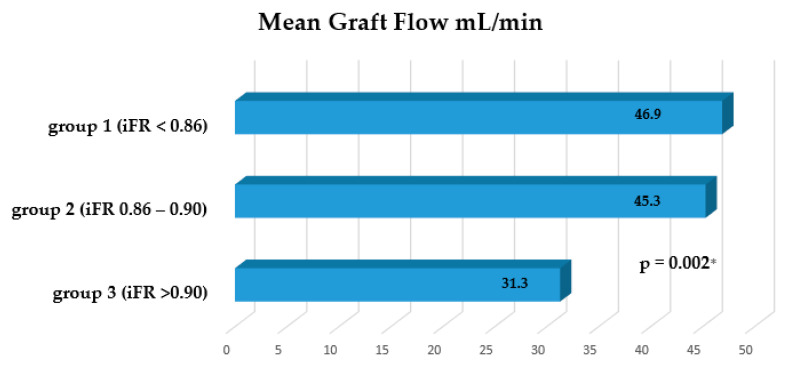
The graph shows a decreasing trend of the mean graft flow when mild stenosis of the coronary artery was confirmed. * Significant difference between group 1 and group 3.

**Figure 4 medicina-56-00714-f004:**
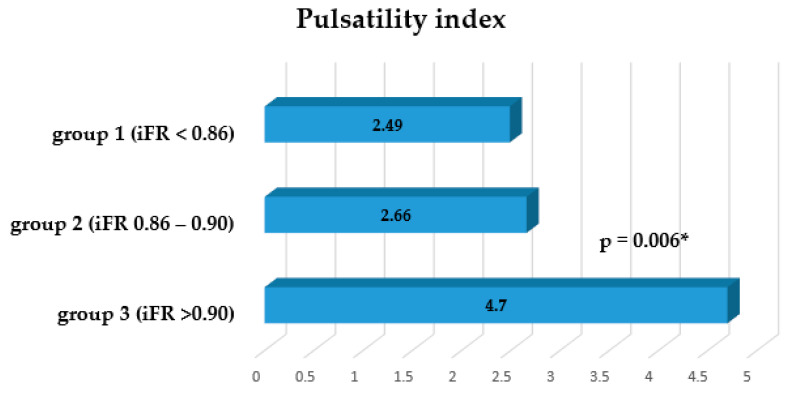
The graph shows a decreasing trend of the pulsatility index data from mild stenosis to significant stenosis of the coronary artery. * Significant difference between group 1 and group 3.

**Figure 5 medicina-56-00714-f005:**
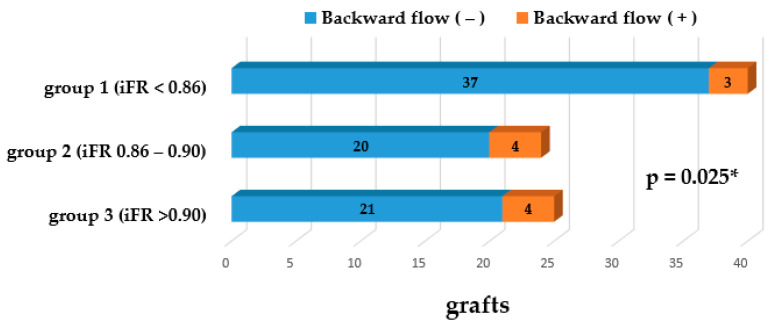
The graph shows a proportion of distribution backward flow among groups. * Significant difference between group 1 and group 2.

**Figure 6 medicina-56-00714-f006:**
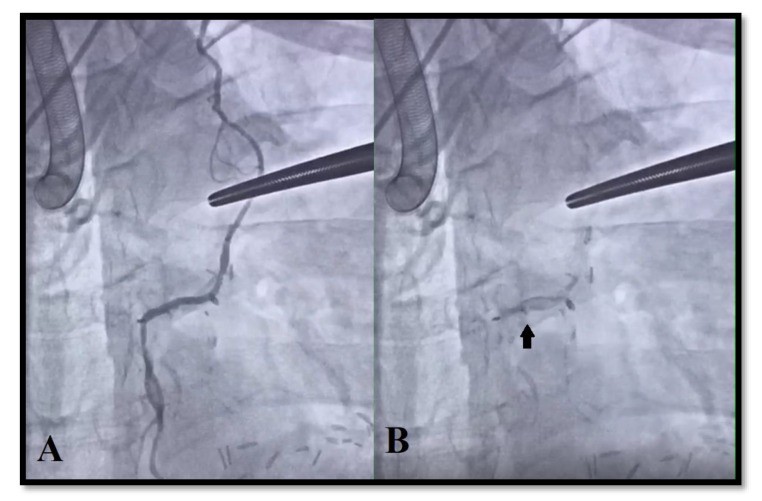
Selective left internal mammary artery (LIMA) graft angiography to the left anterior descending artery (LAD) (**A**). Graft flow: mean graft flow (MGF) 18 mL/min, pulsatility index (PI) 5.1, backward flow (BF) +, diastolic filling (DF) 62%. Reverse contrast flow from left anterior descending artery to left internal mammary artery graft (**B**).

**Table 1 medicina-56-00714-t001:** Patient characteristics.

Variables	Value
Age	63.8 ± 8.9 (48–78)
Body mass index (kg/m^2^)	27.2 ± 4.79
Sex	
Male	23 (92%)
Hypertension	23 (92%)
Diabetes mellitus	2 (8%)
Dyslipidemia	22 (88%)
Arrhythmia	4 (16%)
Previous PCI	9 (36%)
Previous MI	15 (60%)
History of smoking	15 (60%)
LV EF%	47.12 ± 6.4
**NYHA**	
II	19 (76%)
III	6 (24%)
Euro Score II	1.38 ± 0.75
Syntax score	31.38 ± 4.33
Distal anastomoses per 1 pt.	3.56 ± 0.82 (2–5)

Abbreviations: PCI—percutaneous coronary intervention; MI—myocardial infarction; LV EF—left ventricular ejection fraction; NYHA —New York Heart Association; and pt—patient.

**Table 2 medicina-56-00714-t002:** Graft characteristics.

Variables	Group 1 (iFR < 0.86)	Group 2 (iFR 0.86–0.90)	Group 3 (iFR > 0.90)
Distal anastomosis	40	24	25
Grafts	36	22	23
SVG to RCA	7 (17.5%)	4 (16.6%)	3 (12%)
SVG to PDA	8 (20%)	5 (20.8%)	2 (8%)
SVG to OM	11 (27.5%)	6 (25%)	9 (36%)
SVG to diagonal	3 (7.5%)	2 (8%)	4 (16%)
LIMA to LAD	11 (27.5%)	7 (29%)	7 (28%)
Sequential grafts	4	2	2

Abbreviations and explanation: SVG—saphenous vein grafts; RCA—right coronary artery; PDA—posterior descending artery; OM—obtuse marginal artery; LIMA—left internal mammary artery; LAD—left anterior descending artery; total number (percentage of all grafts in a group).

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
