# Peer review of "Correlation between Preoperative Coronary Artery Stenosis Severity Measured by Instantaneous Wave-Free Ratio and Intraoperative Transit Time Flow Measurement of Attached Grafts"

_medicina, 2020, doi:10.3390/medicina56120714_

Round 1

Reviewer 1 Report

The authors have addressed all my comments in a satisfactory fashion.

Author Response

thanks

Reviewer 2 Report

In this work, authors aimed to show if the instantaneous wave-free ratio (iFR), a functional physiological parameter of coronary artery stenosis correlates with intraoperative transit time flow measurement (TTFM) of attached grafts in patients that underwent CABG procedure. Besides, they underwent to see if there is a flow competition between the native coronary artery and the attached graft according to the severity of stenosis. These results are of interest since the only available study to date assessed TTFM concerning the competitive flow of arterial grafts among FFR-guided CABG patients.

The ethical disclosures are valid and provided in the manuscript.

There are some shortcomings that I believe would improve the manuscript, as listed below:

  1. Was the enrollment of 25 patients consecutive? If so, that should be stated in the Methods.
  2. As authors state in the Methods, they divided their patients into three iFR subgroups, one with iFR >0.9, iFR 0.86-0.90 - so-called „gray zone“ and under 0.86 that is considered as a functionally severe coronary stenosis. The authors should substantiate and back these claims with more data. For example, a good paper that studied optimal FFR and iFR thresholds for predicting the physiological significance of CAD is by Modi et al. Circ Cardiovasc Interv. 2018 and should be included.
  3. While I understand that these were patients with stable CAD, I am concerned about the definition and adjudication of the „intermediate“ lesion that is provided in the Methods. Authors claim that angiographically intermediate stenoses comprised those with percent luminal shortening between 50-90% which is quite different than what we normally perceive as intermediate lesions. To most of us, the intermediate lesion would include lesions that are visually from about 40% to 70% on the angiogram. The authors should explain this discrepancy and be more precise. Please refer to work such as Tobis et al. JACC. 2007.
  4. I think that this manuscript would benefit from the „flowchart“ that shows how patients were enrolled in which order was procedures undertaken (e.g. grafting after angiography findings, and iFR measurements after grafting, etc.). This must be clarified.
  5. Did you use the pullback method during the iFR measurement? If so, this should be stated and described in greater detail. Also, were the measurements made by the same operator? This should be elaborated to make sure that the assessment bias was not present.
  6. I would suggest to authors that figures 2 and 3 are shown differently. They should be depicted as bars with mean ± standard deviation and the statistically significant difference between groups should be marked on the figure with an asterisk sign and respective values.
  7. The selection of venous vs. arterial grafts would be more ideal in this scenario and would give us more meaningful data, while this work provides a mix of both since venous and arterial grafts are used, although, authors note that the distribution of venous and arterial grafts was equal between the groups. However, this is a small sample size and this should be regarded as the limitation of the study.
  8. I am surprised that the authors did not try to run a correlation analysis to see if iFR as a functional parameter would correlate with mean graft flow and pulsatility index. I would be interested to see this analysis. It could also be interesting to see if these results would differ if you would make two subgroups and test correlation in each group separately. Groups should be – all arterial grafts and one with all SVGs.
  9. What was the baseline treatment of the patients? Basic pharmacotherapy data at baseline should be shown.
  10. Limitations are well-elaborated.

Author Response

We are very grateful for the work you have done. Your every comment is very important to us. We tried to improve the shortcomings.

Comments and answers:

  1. “Was the enrollment of 25 patients consecutive? If so, that should be stated in the Methods”.

  • Answer: the recruitment of the patients was consecutive. We will include this information in the article (Methods).

  1. “As authors state in the Methods, they divided their patients into three iFR subgroups, one with iFR >0.9, iFR 0.86-0.90 - so-called „gray zone“ and under 0.86 that is considered as a functionally severe coronary stenosis. The authors should substantiate and back these claims with more data. For example, a good paper that studied optimal FFR and iFR thresholds for predicting the physiological significance of CAD is by Modi et al. Circ Cardiovasc Interv. 2018 and should be included.”

  • Answer: we understand the significance and accuracy of the iFR value, especially the cut-off point. Thank you for the suggested article, it will fulfill our data with strong rationale.

  1. “While I understand that these were patients with stable CAD, I am concerned about the definition and adjudication of the „intermediate“lesion that is provided in the Methods. Authors claim that angiographically intermediate stenoses comprised those with percent luminal shortening between 50-90% which is quite different than what we normally perceive as intermediate lesions. To most of us, the intermediate lesion would include lesions that are visually from about 40% to 70% on the angiogram. The authors should explain this discrepancy and be more precise. Please refer to work such as Tobis et al. JACC. 2007.”
  • Answer: in our study intermediate lesion would include lesions that are visually from about 40% to 75% on the angiogram. As bigger stenoses in 90% of cases are hemodynamically significant and there would be a very big risk of plaque rupture by inserting a pressure wire which can lead to PCI instead of CABG.

  1. “I think that this manuscript would benefit from the „flow chart“ that shows how patients were enrolled in which order was procedures undertaken (e.g. grafting after angiography findings, and iFR measurements after grafting, etc.). This must be clarified.”

  • Answer: initially, according to the rules of the Medicina jornal, we were limited by the number of tables, diagrams and figures. But we understand that the flow chart will make it easier to understand the sequence of our research and we have included it.

  1. “Did you use the pullback method during the iFR measurement? If so, this should be stated and described in greater detail. Also, were the measurements made by the same operator? This should be elaborated to make sure that the assessment bias was not present.”
  • Answer: yes, all the physiology measurements were done by the same operator and in all targeted vessels iFR was measured distally and with pullback to localize the most severe lesion. Will include this information.
  1. “I would suggest to authors that figures 2 and 3 are shown differently. They should be depicted as bars with mean ± standard deviation and the statistically significant difference between groups should be marked on the figure with an asterisk sign and respective values.”

  • Answer: we replaced charts with bars for better visualization

  1. “The selection of venous vs. arterial grafts would be more ideal in this scenario and would give us more meaningful data, while this work provides a mix of both since venous and arterial grafts are used, although, authors note that the distribution of venous and arterial grafts was equal between the groups. However, this is a small sample size and this should be regarded as the limitation of the study.”

  • Answer: we understand that it would be ideal to separate the venous and arterial grafts separately, also according to the choice of the area of revascularization. For our study, of course, the power of analysis will be weakened with this division. This study combines the cardiologists and surgeons’ efforts, and recruiting a large sample of patients within a certain period of time is a limitation of our study.

  1. “I am surprised that the authors did not try to run a correlation analysis to see if iFR as a functional parameter would correlate with mean graft flow and pulsatility index. I would be interested to see this analysis. It could also be interesting to see if these results would differ if you would make two subgroups and test correlation in each group separately. Groups should be – all arterial grafts and one with all SVGs”.

  • Answer: within the framework of this study, we focused on data on the change in graft flow depending on the degree of coronary artery lesion. We have previously performed a correlation analysis. The correlation coefficient between iFR and MGF was -0.372 (p = 0.024), between iFR and PI was 0.428 (p = 0.044) in all grafts. In venous grafts, separately, the correlation coefficient between iFR and MGF was -0.330 (p = 0.064), between iFR and PI was 0.275 (p = 0.091). In arterial grafts, the coefficient of correlation between iFR and MGF was -0.460 (p = 0.048), between iFR and PI was 0.563 (p = 0.002).

  1. “What was the baseline treatment of the patients? Basic pharmacotherapy data at baseline should be shown”

  • All the patients were on a standard treatment according to ECS guidelines of Chronic coronary syndrome (Aspirin, BAB, ACEI, Statin). Will include this information.

Knuuti, J., et. al, 2019 ESC Guidelines for the diagnosis and management of chronic coronary syndromes: the Task Force for the diagnosis and management of chronic coronary syndromes of the European Society of Cardiology (ESC). European heart journal, 41(3), pp.407-477.

  1. “Limitations are well-elaborated”

  • Answer: thank you.